# Factors influencing the bias between blood gas analysis versus central laboratory hemoglobin testing. A secondary analysis of a randomized controlled trial

Linda Tanner[1], Simone Lindau[1], Markus Velten[2], Tobias Schlesinger[3], Maria Wittmann[2], Peter Kranke[3], Kira Berg[1], Florian Piekarski[1], Christoph Füllenbach[1], Suma Choorapoikayil[1], Dirk Hasenclever[4], Kai Zacharowski[1], Patrick Meybohm[1,3]*

1 Department of Anesthesiology, Intensive Care Medicine and Pain Therapy, University Hospital Frankfurt, Goethe University Frankfurt, Frankfurt am Main, Germany, 2 Department of Anesthesiology and Operative Intensive Care Medicine, University Hospital Bonn, Rheinische Friedrich-Wilhelms-University Bonn, Bonn Germany, 3 Department of Anesthesiology, University Hospital Wuerzburg, Wuerzburg, Germany, 4 Institute for Medical Informatics, Statistics and Epidemiology (IMISE), Medical Faculty of the University Leipzig, Leipzig, Germany

* Meybohm_P@ukw.de

**Data Availability Statement:** Access restrictions apply to the data underlying the findings. Ethical restrictions prevent public sharing of data since the LIBERAL Trial still is a recruiting randomized

## Abstract

### Background

Anemia is the most important complication during major surgery and transfusion of red blood cells is the mainstay to compensate for life threating blood loss. Therefore, accurate measurement of hemoglobin (Hb) concentration should be provided in real-time. Blood Gas Analysis (BGA) provides rapid point-of-care assessment using smaller sampling tubes compared to central laboratory (CL) services.

### Objective

This study aimed to investigate the accuracy of BGA hemoglobin testing as compared to CL services.

### Methods

Data of the ongoing LIBERAL-Trial (Liberal transfusion strategy to prevent mortality and anemia-associated ischemic events in elderly non-cardiac surgical patients, LIBERAL) was used to assess the bias for Hb level measured by BGA devices (ABL800 Flex analyzer®, GEM series® and RapidPoint 500®) and CL as the reference method. For that, we analyzed pairs of Hb level measured by CL and BGA within two hours. Furthermore, the impact of various confounding factors including age, gender, BMI, smoker status, transfusion of RBC, intraoperative hemodilution, and co-medication was elucidated. In order to ensure adequate statistical analysis, only data of participating centers providing more than 200 Hb pairs were used.

controlled trial and the database has not been finalized. Thus, it is protected against unauthorized access. Access can be granted to authorized persons by the sponsor. Data can be obtained by contacting the Clinical Trial Centre Leipzig, University of Leipzig, Härtelstraße 16-19, 04107 Leipzig (info@zks.uni-leipzig.de).

**Funding:** This work is supported by the German Research Foundation (Deutsche Forschungsgemeinschaft; grant no. ME 3559/3-1 to PM) who had no impact in the design of the study and collection, analysis, and interpretation of data and in writing the manuscript. https://www.dfg.de/en/index.jsp.

**Competing interests:** The authors have declared that no competing interests exist.

## Results

In total, three centers including 963 patients with 1,814 pairs of Hb measurements were analyzed. Mean bias was comparable between ABL800 Flex analyzer® and GEM series®: -0.38 ± 0.15 g/dl whereas RapidPoint 500® showed a smaller bias (-0.09 g/dl) but greater median absolute deviation (± 0.45 g/dl). In order to avoid interference with different standard deviations caused by the different analytic devices, we focused on two centers using the same BGA technique (309 patients and 1,570 Hb pairs). A Bland-Altman analysis and LOW-ESS curve showed that bias decreased with smaller Hb values in absolute numbers but increased relatively. The smoker status showed the greatest reduction in bias (0.1 g/dl, p<0.001) whereas BMI (0.07 g/dl, p = 0.0178), RBC transfusion (0.06 g/dl, p<0.001), statins (0.04 g/dl, p<0.05) and beta blocker (0.03 g/dl, p = 0.02) showed a slight effect on bias. Intraoperative substitution of volume and other co-medications did not influence the bias significantly.

## Conclusion

Many interventions like substitution of fluids, coagulating factors or RBC units rely on the accuracy of laboratory measurement devices. Although BGA Hb testing showed a consistently stable difference to CL, our data confirm that BGA devices are associated with different bias. Therefore, we suggest that hospitals assess their individual bias before implementing BGA as valid and stable supplement to CL. However, based on the finding that bias decreased with smaller Hb values, which in turn are used for transfusion decision, we expect no unnecessary or delayed RBC transfusion, and no major impact on the LIBERAL trial performance.

## Introduction

Acute anemia is one of the most important and common complication during and after surgery [1]. While surgical techniques have advanced over the course of time, intraoperative blood loss is still present in major surgery. Even though infusions or coagulating factors are often administrated to compensate massive blood loss, transfusion of allogenic red blood cells (RBC) is in many cases inevitable [2]. In these critical situations hemoglobin (Hb) and/or hematocrit levels are one key factor among others to determine the need for RBC transfusion. It is noteworthy to mention that transfusion guidelines also recommend to consider physiological constitution, hemodynamic and spirometry parameters, volume status, and dynamic of bleeding.

Several methods, such as central laboratory (CL), blood gas analysis (BGA) or CO-Oximetry [3] are applied to evaluate the Hb level, of which BGA and CL are the most common used methods intraoperatively. However, measurement values differ between BGA and CL [4–7]. Studies compared different devices in order to establish the optimal, less invasive, fastest but at the same time most accurate method to measure Hb level. For example, Giraud et al compared the accuracy of four different bedside devices with CL. The analysis of 219 measurements from 53 patients revealed different Hb levels for each device. Among all methods, the point-of-care testing (POCT) device HemoCue® displayed the smallest and CO-Oximetry the biggest bias [3]. Non-invasive Hb measurement, such as spectrophotometry, turned out to be inferior to invasive Hb measurement (BGA) [8]. Additionally, studies investigated whether differences in

clinical routines or patient's physiological status affect Hb values using different devices for measurements. For example, the concentration of Hb varies depending on the amount of plasma volume, leading to different Hb levels between arterial or venous samples within the same patient. In addition, position of the body or time of the day during blood withdrawal also appear to influence Hb level [9]. In cases of hemodilution with hematocrit below 30%, measurement of Hb concentration using CO-Oximetry seems to be more accurate than conductivity [10]. Ng and colleagues evaluated the bias in Hb level between CL and POCT device before and after major blood loss ($\geq$ 25% of blood volume). The analysis revealed that Hb levels were lower when measured with POCT device as compared to laboratory. Therefore, the authors recommend physicians not to use the POCT device in particular situation such as extensive blood loss when accurate measurement is essential [11]. To the best of our knowledge, however, the bias between BGA measurement and CL depending on the patient's physiology, co-medication and clinical events such as blood transfusion or intraoperative volume therapy has not been assessed. Additionally, accurate measurement of Hb level is important for studies ultimately implementing clinical guidelines. A number of studies compared surgical outcome in patients receiving either a restrictive or liberal transfusion regime. In these studies, several methods have been used to detect Hb concentration, however details about the used method for measurement are often not described. In addition, it is not evident whether the same methods were used in multicenter studies or more importantly considered during the analysis [12–15].

Here, we performed a sub-analysis of the ongoing "liberal transfusion strategy to prevent mortality and anemia-associated ischemic events in elderly non-cardiac surgical patients" (LIBERAL-Trial) [16] to investigate the accuracy of BGA Hb testing compared to CL by determining factors that potentially influence measurements in surgical patients.

## Material and methods

### Study design

This sub-analysis was conducted with data from the LIBERAL-Trial (NCT03369210) [16]. The LIBERAL-Trial is a prospective, open, multicenter, randomized phase IV clinical trial to investigate whether a liberal transfusion strategy of RBCs prevents mortality and anemia-associated ischemic events in elderly patients undergoing non-cardiac major surgery. Briefly, elderly patients ($\geq$70 years) scheduled for intermediate or high risk non-cardiac surgery were included in the trial and randomized to a restrictive or liberal transfusion group as soon as Hb level dropped below $\leq$9 g/dl during surgery or postoperative day 1 to 3. Hemoglobin was measured intraoperatively and on daily basis. The LIBERAL-Trial was approved by the Ethics Committee of the University of Frankfurt (Ref: 139/17F) and by the federal authority (Paul-Ehrlich-Institute) [16].

### Blood sampling

During hospital stay, Hb measurement by CL or BGA occurred regularly. Pairs of Hb were taken in the recovery room after surgery. Furthermore, if the patient remained for 24 hours in the recovery room, another pair was taken the following day at 4 am. If patients were admitted to the intensive care unit or intermediate care unit, pairs of Hb were taken at 4am and 4pm on a regular basis. As stated in the study protocol [16] during massive bleeding or before RBC transfusion, additional blood samples were analysed using CL or BGA.

### Central laboratory measurements

Hemoglobin level was determined with SYSMEX XN-10® as part of the SYSMEX XN-9000® using the Sodium-Lauryl-Sulfate (SLS) method in the local CL department of all three trial

sites. EDTA tubes with either arterial or venous whole blood were used for blood analysis. Analysis with CL was mainly used at the peripheral ward during pre- and postoperative management. Results were displayed approximately within 20 or 60 minutes on weekdays and weekends, respectively.

## Blood gas analysis

To determine the Hb level of either arterial or venous heparinized blood samples, the ABL800 Flex®️ analyzer was used at the LIB-07 center and the GEM Series®️ at the LIB-26 center. In the LIB-05 center, the RapidPoint 500®️ was used to measure the Hb level using CO-oximetry. Hemoglobin level was quantified with using the law of Lambert-Beersche. If Hb was <0.16 g/dl (0.1 mmol/l) or >40.26 g/dl (25 mmol/l), results were classified as outlier and not considered in the analysis. Blood samples, both arterial and venous, were taken by trained medical staff and transferred to the BGA analyzer immediately. Results were displayed in the patients electronic medical file within 65 seconds.

## Data collection

Blood Gas Analysis and CL measurements are subject to internal and external quality control according to the guidelines of the German Medical Association guidelines in medical laboratory examinations [17]. Data of patients enrolled in the LIBERAL-Trial from January 2018 until October 2019 were extracted and analyzed. Hemoglobin measurements were collected before, during and after surgery until hospital discharge or 30 days postoperative, whichever occurred first [16]. For estimation of intraoperative or postoperative anemia according to the definition of the WHO [18], both Hb measurements through BGA devices and CL were used. We did not discriminate between arterial or venous blood sample as the patient's electronic case report file does not provide information on sample's origin. Blood sampling occurred at least every third day. After RBC transfusion, Hb concentration was examined to ensure that the target Hb level had been reached within 24 hours. Several patient characteristics were investigated including Body Mass Index (BMI), gender, 15 most common used co-medication (ACE-Inhibitor (ACE), beta blocker (BETA), calcium channel blockers (CAANT), aspirin (ASS), benzodiazepine (BENZO), statins (STAT), insulin (INS), oral antidiabetic drugs (ODIA), antiarrhythmic agents (AARRH), Parkinson medication (PARK), neuroleptics (NEURO), nonsteroidal anti-inflammatory drugs (NSAR), opioids (OPIO), oral anticoagulants (ACOA) and dual platelet aggregation inhibitor (DPLAT)), volume therapy and smoker status. Volume therapy refers to the administration of crystalloid fluids during surgery. Often, 500mL to 1000mL crystalloid solutions are used. Immediately before surgery, crystalloids are applied to resuscitate the deficit in volume from fasting. During the surgery, crystalloid solutions are applied in order to resuscitate the deficit in blood volume from perspiration or bleeding by using physiological marker such as lactate, blood pressure and heart rate. Blood loss and the amount of crystalloids were not measured. Because volume therapy is part of our perioperative guidelines, we compared Hb pairs after volume therapy with postoperative Hb pairs. In order to investigate whether RBC transfusion affects the bias of Hb level measured with BGA or CL, Hb pairs of patients taken within 24 hours after RBC transfusion were compared with non-transfused patients or with patients in which transfusion occurred more than 24 hours prior to Hb measurement. This timeframe was used because an increase in Hb level is expected within two to 24 hours after RBC transfusion [17].

## Statistical analysis

We analyzed pairs of CL and BGA measurements. A pair was defined as two Hb values obtained within 2 hours from CL and BGA. To ensure robust analysis we included only centers

with n >200 Hb pairs. Data are presented as mean values with standard deviation. Robust statistical methods such as median absolute deviation (MAD) were used to assess bias for extreme outliers, for example in cases of RBC transfusion or massive blood loss during the two hours time range. Outliners were excluded if bias displayed a value 5 times of standard deviation (SD) of the respective median value. Statistical significance level was accepted with p <0.05. Bland-Altman analysis was performed to analyze the agreement between CL and BGA measurements [19]. LOWESS curve was used to describe the development of bias in decreasing Hb values. Multiple regression analysis and t-test were applied to determine significance of the bias depending on various factors including patient characteristics (BMI, gender, 15 most common used co-medication and smoker status), RBC transfusion and volume substitution. The results are displayed using empirical cumulative distribution functions. Logarithmic scales were used to meet the range of Hb values. Except for center and Hb-level, analysis of several other factors is strictly explorative. Because we expected no differences to our results, we did not adjust for multiplicity. R-Development Core Team (2008), Version 3.6.1, R Foundation for Statistical Computing, Vienna, Austria) was used for statistical analysis.

## Results

In total, three centers including 963 patients with 1,814 pairs of Hb measurements were analyzed. The mean difference in Hb pairs of CL and BGA measurements taken within two hours was assessed to determine the biases for center LIB-07 (n = 1307), center LIB-26 (n = 263) and center LIB-05 (n = 244), respectively. Mean bias was comparable between LIB-07 and LIB-26. Empirical cumulative distribution function showed that approximately 95% of all CL values are smaller than BGA (Fig 1). LIB-07 and LIB-26 revealed a mean bias of -0.38 ± 0.28 and

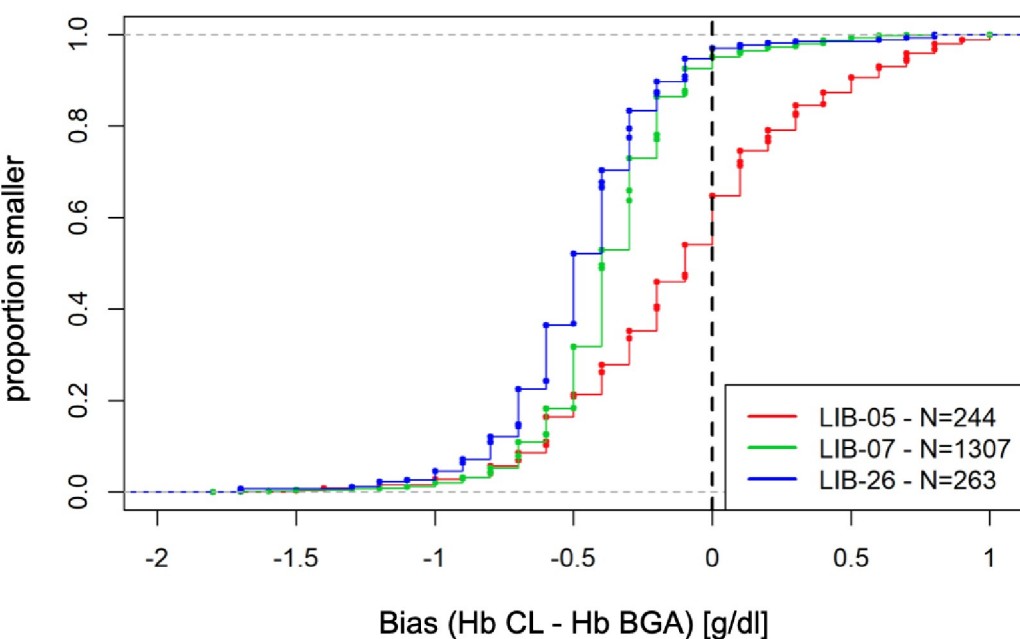

**Fig 1. Comparison of the bias.** Empirical cumulative distribution function was applied to detect potential bias between centers (LIB-05, LIB-07 and LIB-26). Hb = hemoglobin, CL central lab, BGA = blood gas analysis, LIB-05 = Bonn, LIB-07 = Lrankfurt, LIB-26 = Wuerzburg.

MAD ± 0.15 g/dl (median -0.4 g/dl) (Fig 2A) whereas LIB-05 showed a smaller bias (-0.09 ± 0.45 g/dl) but greater MAD (± 0.45 g/dl) (median -0.1) (Fig 2B).

In order to avoid interference with different standard deviations from the trial sites that may be caused by the different analytic devices, we focused on two centers using the same BGA technique: LIB-07 and LIB-26. In total 309 patients and 1,570 Hb pairs were included in the further analysis. Overall, 1,389 Hb pairs were measured ≥8 g/dl and 184 Hb pairs between 6 and 8 g/dl. To assess a relation between the difference in each pair of values and the respective mean Hb value, data pairs were plotted using a Bland-Altman Plot (Fig 3). The LOWESS curve for absolute Hb values showed that low Hb values are associated with smaller bias. Hb values of 10 to 15 g/dl display a bias with -0.5 g/dl whereas Hb values of 6 g/dl display a bias of -0.25 g/dl (Fig 3). However, Hb differences plotted as percentage revealed an increase of up to 5% of the Hb value that are associated with smaller Hb values (Fig 4).

### Impact of various patient characteristics on bias

We compared Hb pairs to investigate whether various patient characteristics influence the bias between BGA and CL measurement. Age, gender or volume therapy during surgery did not significantly influence bias (Table 1). Significant differences in mean bias were detected for BMI, smoker status and for patients with and without transfusion. Mean bias was reduced by 0.07 g/dl in underweight (-0.29 ± 0.33 g/dl, BMI < 19) compared to overweight patients (-0.36 ± 0.28 g/dl, BMI > 26) (p = 0.0178) (Table 1) and reduced by 0.1 g/dl in smokers (-0.30 ± 0.24 g/dl) compared to non-smoker (-0.40 ± 0.29 g/dl) (p<0.001). Mean bias between ex-smoker (-0.37 ± 0.28 g/dl) and non-smoker was similar (-0.40 ± 0.29 g/dl) (Table 1, Fig 5). Among 309 patients, 141 received at least transfusion of one unit of RBC and/or autologous blood. Mean bias was reduced by 0.06 g/dl in patients after RBC transfusion (-0.34 ± 0.26 g/dl) compared to patients without transfusion (-0.4 ± 0.29 g/dl) (p<0.001) (Table 1).

### Intraoperative volume therapy versus postoperative values

To elucidate whether intraoperative volume substitution with crystalloid fluids affects the bias, we compared blood samples obtained immediately after surgery in the recovery room (n = 271) with samples taken up to 24 hours after skin incision (postoperative, n = 13). Mean bias was non-significantly reduced by 0.02 g/dl in patients after 24 hours of volume substitution (-0.37 ± 0.29 g/dl) compared to values immediately after surgery (-0.39 ± 0.26 g/dl) (p = 0.202) (Table 1).

### Impact of co-medications on bias

Several co-medications (ACE, BETA, CAANT, ASS, BENZO, STAT, INS, ODIA, AARRH, PARK, NEURO, NSAR, OPIO, ACOA, and DPLAT) where taken by the patients at hospital admission. BETA and STAT influenced the bias in Hb level significantly (Table 2). Mean bias was reduced by 0.03 g/dl for patients with betablockers (-0.4 ± 0.28 g/dl) compared to patients without (-0.37 ± 0.28 g/dl) betablockers (BETA) (p = 0.02) and by 0.04 g/dl for patients with statins (-0.41 ± 0.28 g/dl) compared to patients without (-0.37 ± 0.28 g/dl) statins (STAT) (p<0.05) (Table 2).

### Discussion

Measurement of diagnostic parameters using BGA is an integral part of clinical routine providing fast results and enabling rapid response in critical situations. However, results of BGA devices need to be regarded cautiously due to conflicting results from various studies. Many

(median absolute deviation), Hb= hemoglobin, CL = central lab, BGA = blood gas analysis, LIB-05 = Bonn, LIB-07 = Frankfurt, LIB-26 = Wuerzburg

## Histogram with Density

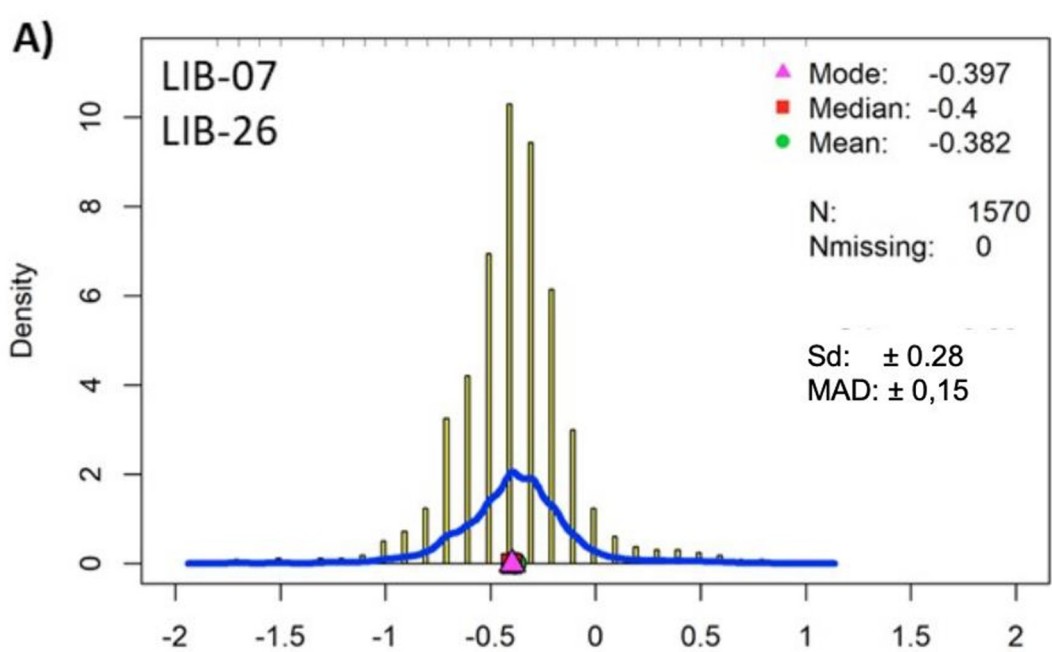

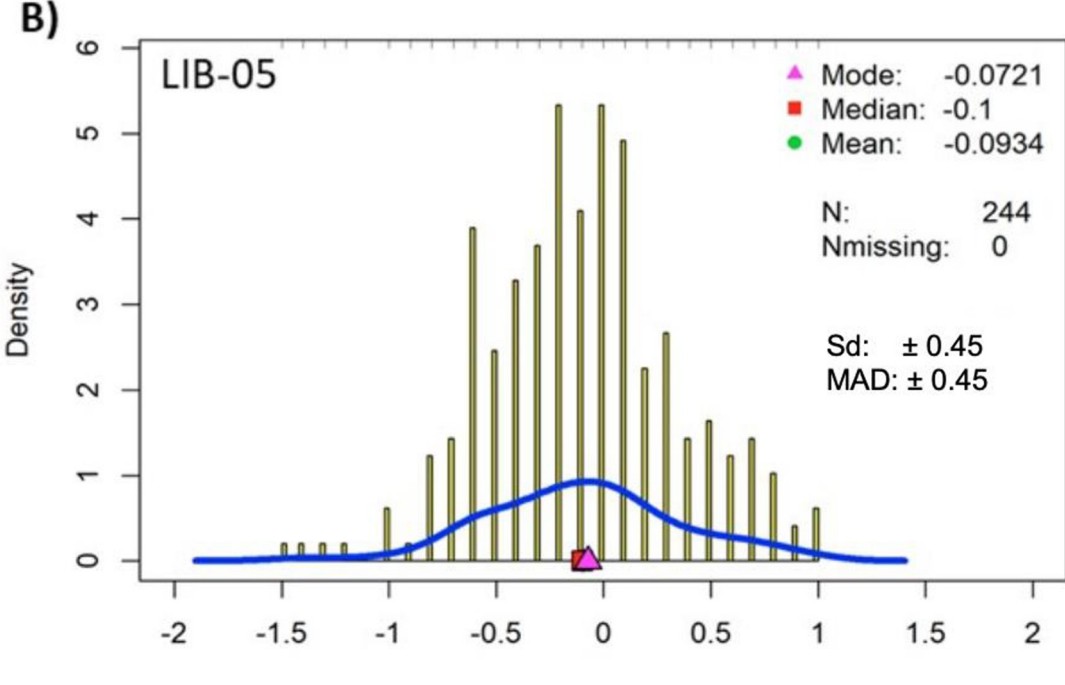

**Fig 2. A** and B: Histogram with density for hemoglobin pairs from LIB 07 and LIB-26. The relative frequency of Hb difference is displayed (a, b). Sd (Standard deviation), MAD.

interventions like substitution of fluids or coagulating factors rely on the accuracy of laboratory measurement devices [20]. When looking at clinically relevant decisions, for example within randomized trials comparing outcome of different transfusion strategies, accurate measurement of Hb values should be ensured anytime [13–15]. Here, we investigated the accuracy of BGA compared to CL by determining factors that can potentially influence these measurements. In total, 1,814 pairs of Hb measurements of 963 patients of the LIBERAL-Trial have been investigated of which 1,570 pairs of 309 patients were analyzed in detail. Interestingly, the greatest influence on the bias results from different BGA devices used by the trial sites. Mean bias is comparable between ABL800 Flex® analyzer and the GEM series® (-0.38 ± 0.28 g/dl) whereas RapidPoint 500® showed smaller bias but greater MAD. Overall, the Bland-Altman analysis and LOWESS curve revealed that bias decreased with smaller Hb values. We also examined several factors that could lead to bias in Hb values. Of all investigated factors smoker status showed the greatest effect whereas BMI, RBC transfusion, BETA and STAT showed only a slight effect on bias. Bias was reduced by 0.1 g/dl in smokers compared to non-smoker. This might be caused by the different methods used to measure Hb concentration. The SLS-method used by CL is sensitive to methemoglobin, which is increasingly found in smokers and

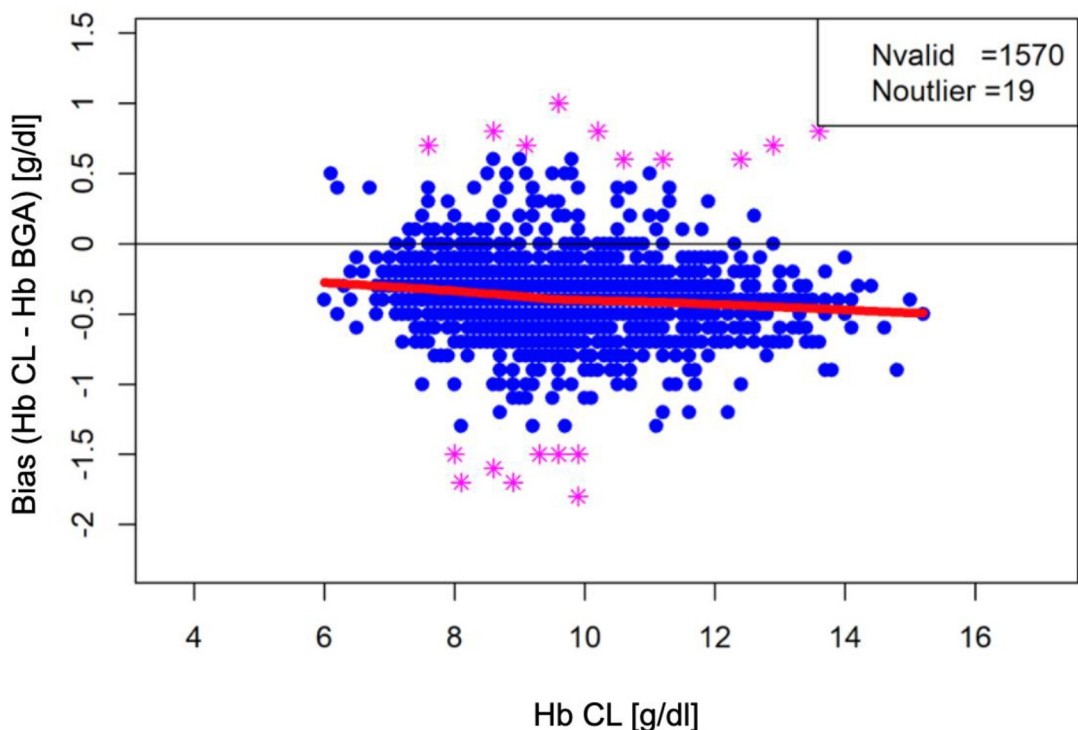

**Fig 3. Bland-Altman Plot and LOWESS curve showing the differences of the Hb differences (LIB-07 and LIB-26).** The difference of the Hb values (Hb-CL and Hb-BGA) calculated for each pair and plotted against the mean value of both measurements are displayed ((Hb-CL + Hb-BGA)/2). The LOWESS curve (red line) shows the tendency of mean bias with lower Hb levels. Hb = hemoglobin, CL = central lab, BGA = blood gas analysis.

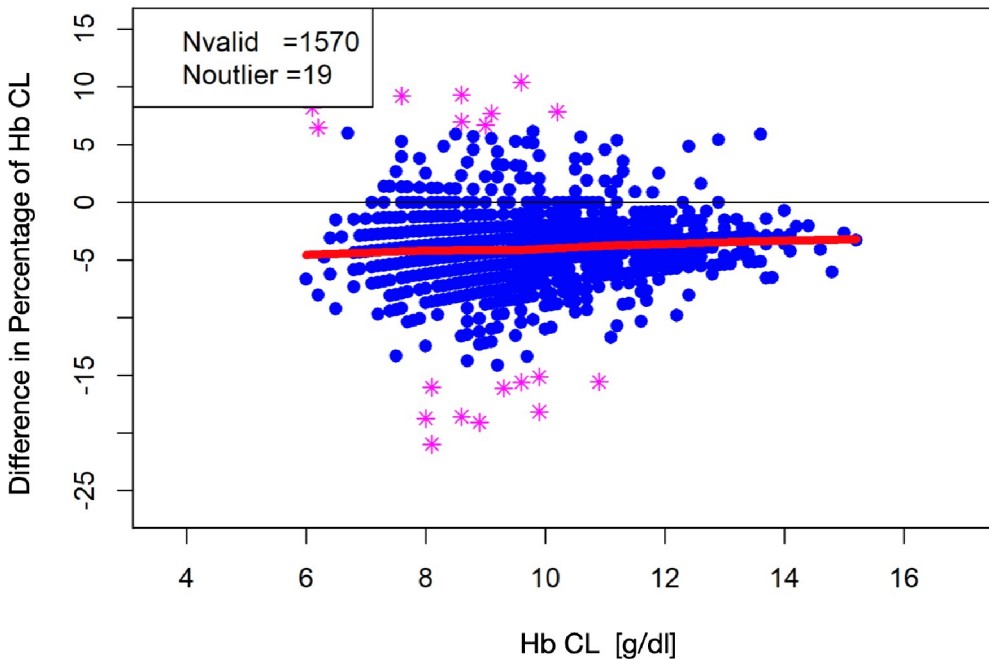

**Fig 4. Bland-Altman Plot and LOWESS curve showing differences in percentage of Hb pairs (LIB-07 and LIB-26).**
The difference in percentage of Hb values (Hb-CL and Hb- BGA) calculated for each pair and plotted against the mean value of both measurements are displayed ((Hb-CL + Hb-BGA)/2). The LOWESS curve (red line) shows the tendency of mean bias expressed as percentage of CL Hb with lower Hb levels. Hb = hemoglobin, CL = central lab, BGA = blood gas analysis.

may result in falsely higher concentration of Hb measured by CL [21]. Furthermore, we observed that transfusion of RBC slightly reduced the difference in bias by 0.06 g/dl. Similarly, we found slight differences of 0.07 g/dl for BMI (underweight vs overweight), 0.03 g/dl for

**Table 1. Possible confounding factors N = absolute number of patients, SD = standard deviation (LIB-07 and LIB-26).**

| Variable | Description | N | Mean Bias (g/dl) | SD Bias (g/dl) | P-value |
|---|---|---|---|---|---|
| Age | < 80 years | 224 | -0.38 | 0.29 | 0.952 |
| | > 80 years | 85 | -0.38 | 0.27 | |
| Gender | Male | 196 | -0.38 | 0.27 | 0.935 |
| | Female | 113 | -0.38 | 0.29 | |
| BMI | Underweight | 11 | -0.29 | 0.33 | 0.0178 |
| | Normalweight | 132 | -0.37 | 0.27 | |
| | Overweight | 104 | -0.36 | 0.28 | |
| | Obesity class I | 46 | -0.36 | 0.27 | |
| | Obesity class II/III | 16 | -0.45 | 0.32 | |
| Smoker status | Smoker | 33 | -0.30 | 0.24 | 0.000016 |
| | Ex-smoker | 135 | -0.37 | 0.28 | |
| | Non-smoker | 141 | -0.40 | 0.29 | |
| Volume therapy | Intraoperative | 271 | -0.39 | 0.26 | 0.202 |
| | Postoperative | 13 | -0.37 | 0.29 | |
| After RBC transfusion | No transfusion last 24h | 168 | -0.40 | 0.29 | 0.000869 |
| | After transfusion | 141 | -0.34 | 0.26 | |

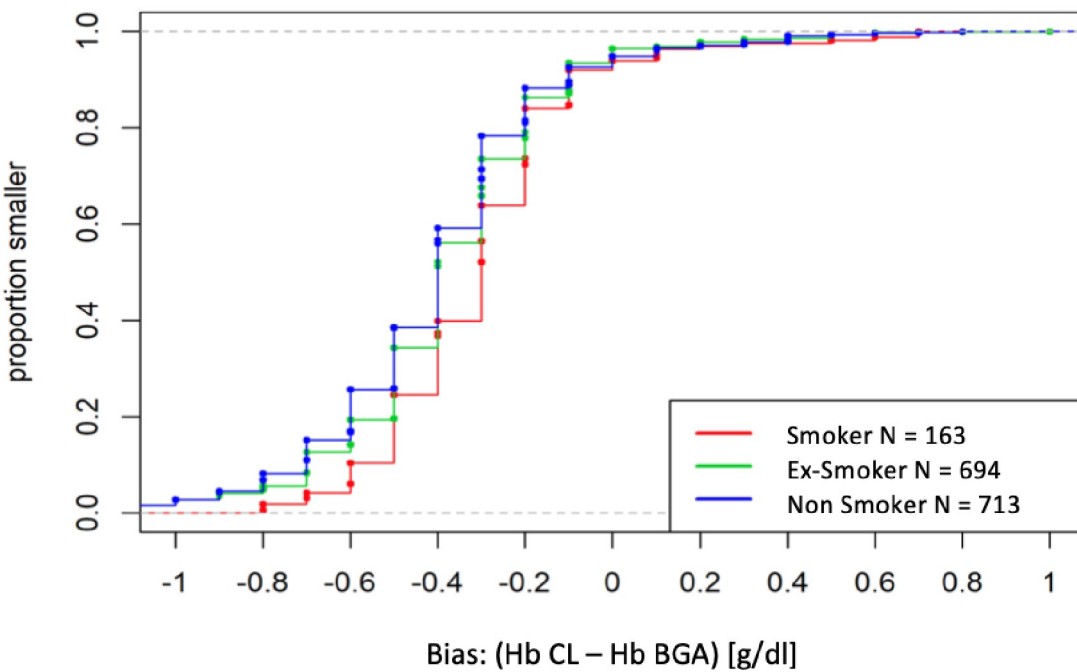

**Fig 5. Bias depending on smoker status (LIB-07 and LIB-26).** The relative frequencies of Hb pairs in smoker, non-smoker and ex-smoker are displayed. Hb = hemoglobin, CL = central lab, BGA = blood gas analysis.

BETA and 0.04 g/dl for STAT. It is noteworthy, that intraoperative substitution of volume and co-medications, except BETA and STAT, did not influence the bias significantly. Based on the stable bias with small standard deviation which were largely unaffected by various physiological factors both BGA devices—ABL800 Flex® analyzer and the GEM series®- ensure comparable and accurate estimation of Hb levels.

Our analysis confirms that BGA produces stable values for Hb and is not influenced by the patient's physiology except for smoker. This observation is especially relevant during trauma setting where no information about the critical patient is available. The bias of smoker is only minor and therefore, we believe, negligible in this setting Hemoglobin measurement is not only important for categorizing the patients' trauma but is also used in several scoring systems, such as the Trauma Associated Severe Hemorrhage Score (TASH) [22]. Prior to RBC transfusion, the physician evaluates if there are possible transfusion triggers such as low Hb values, hypotension, tachycardia or lactatemia. In these settings, particularly in intensive care units where Hb is frequently determined [23,24], measurements with BGA is recommended since less blood volume is required, turn-around time is faster, and stable results for Hb can be observed. Blood sparing methods consequently lead to fewer RBC transfusions and shorter duration of stay in the intensive care unit [23]. Unfortunately, noninvasive methods tended to be inferior to invasive BGA [8,24]. A, faster turn-around time consequently leads to faster decision making. However, once the laboratory results become available, the treatment should be checked and if necessary adjusted accordingly.

In our analysis, we used data of a large prospective multicenter trial where patients are randomized to a restrictive or liberal transfusion group with a target range for post-transfusion Hb concentration of 7.5–9 or 9–10.5 g/dl, respectively [16]. Our analysis show that BGA is a

**Table 2. Biases depending on co-medication (LIB-07 and LIB-26).**

| Medication | | N | Mean Bias (g/dl) | SD Bias (g/dl) | P-Value |
|---|---|---|---|---|---|
| AARRH | no | 298 | -0.38 | 0.28 | 0.689 |
| | yes | 11 | -0.37 | 0.25 | |
| ACE | no | 119 | -0.37 | 0.29 | 0,16 |
| | yes | 190 | -0.39 | 0.28 | |
| ACOA | no | 240 | -0.38 | 0.28 | 0.319 |
| | yes | 69 | -0.39 | 0.29 | |
| ASS | no | 196 | -0.38 | 0.29 | 0.453 |
| | yes | 113 | -0.39 | 0.27 | |
| BENZO | no | 303 | -0.38 | 0.28 | 0.691 |
| | yes | 6 | -0.41 | 0.31 | |
| BETA | no | 151 | -0.37 | 0.28 | 0.0234 |
| | yes | 158 | -0.40 | 0.28 | |
| CAANT | no | 227 | -0.38 | 0.28 | 0.733 |
| | yes | 82 | -0.39 | 0.29 | |
| DPLAT | no | 269 | -0.38 | 0.28 | 0.85 |
| | yes | 13 | -0.38 | 0.26 | |
| INS | no | 277 | -0.38 | 0.28 | 0.485 |
| | yes | 32 | -0.36 | 0.32 | |
| Neuro | no | 295 | -0.38 | 0.28 | 0.823 |
| | yes | 14 | -0.37 | 0.29 | |
| NSAR | no | 287 | -0.38 | 0.28 | 0.704 |
| | yes | 22 | -0.37 | 0.27 | |
| ODIA | no | 262 | -0.38 | 0.28 | 0.762 |
| | yes | 47 | -0.39 | 0.31 | |
| OPIO | no | 267 | -0.38 | 0.29 | 0.44 |
| | yes | 42 | -0.37 | 0.22 | |
| PARK | no | 301 | -0.38 | 0.28 | 0.299 |
| | yes | 8 | -0.43 | 0.25 | |
| STAT | no | 168 | -0.37 | 0.28 | 0.00762 |
| | yes | 141 | -0.41 | 0.28 | |

ACE = ACE-Inhibitor, BETA = beta blocker, CAANT = calcium channel blockers, ASS = aspirin, BENZO = benzodiazepine, STAT = statins, INS = insulin, ODIA = oral antidiabetic drugs, AARRH = antiarrhythmic agents, PARK = Parkinson medication, NEURO = neuroleptics, NSAR = nonsteroidal anti-inflammatory drugs, OPIO = opioids, ACOA = oral anticoagulants, DPLAT = dual platelet aggregation inhibitor, N = absolute number of patients, SD = standard deviation.

stable supplement to CL to measure Hb values. In this sub-analysis of the LIBERAL trial, we found that Hb values measured by BGA vary between different centers. To allow a precise analysis each participating center should determine their individual correction factor which will be applied during our final analysis. This study illustrates the importance of determining the concordance between values obtained by BGA and those obtained in the CL for each individual hospital.

One limitation of our analysis is that we were not able to compare pairs of one blood sample. This may result in a different bias due to dynamic fluid and blood shifts, such as massive blood loss or substitution of crystalloid fluids. However, we addressed this possibility in our analysis and excluded outliners by using robust statistical methods: bias deviated only slightly

from the median (MAD ± 0.15 g/dl) and thus can be regarded as stable. Here we compared the ABL800 Flex® analyzer, GEM series® and RapidPoint 500®, therefore, our results do not apply to other measurement methods used within clinical routine. Another limitation is the multiple testing we used for this analysis. Ideally, equal number of Hb pairs for the different ranges of ≤6, 6–8, and ≥8 g/dl would be preferred for analysis. However, our observed distribution of higher amount of Hb pairs for 8 g/dl which are followed by 6–8 g/dl represent the clinical distribution of Hb values of patients undergoing non-emergent major surgical procedures. Furthermore, we did not analyze the bias of pre- or intraoperative Hb pairs, which could have contributed to the evaluation of bias during massive bleeding. Finally, due to the study protocol we were not able to differentiate between arterial or venous samplings which may also affect the bias due to possible differences in plasma volume. Since BGA is frequently used in clinical practice to assess Hb status and potentially trigger clinical decisions, we suggest that future trials should consistently use either venous or arterial blood for analysis.

Taken together, we investigated the accuracy of three BGA devices to measure Hb concentration with CL as reference method within the ongoing multicenter randomized controlled LIBERAL-Trial. Our analysis revealed that BGA devices used within the trial are associated with different biases. Of all investigated possible confounders only smoker status was systematically related to bias. Multicenter trials, such as the LIBERAL-Trial, should assess whether transfusion decision was based on Hb concentration estimated with BGA or CL. Our analysis showed that bias increases relatively with lower Hb values. However, clinically these findings are minimal (0.25 g/dl for Hb pairs from 6 to 8 g/dl). This Hb range is used for transfusion decision and we expect no unnecessary or delayed RBC transfusion, and no major impact on the LIBERAL-Trial performance. A small bias is negligible, and clinicians should not hesitate to trust measurements using BGA devices. Nevertheless, we suggest that hospitals assess their individual bias before implementing BGA as valid and stable supplement to CL.

## Author Contributions

**Conceptualization:** Linda Tanner, Dirk Hasenclever, Kai Zacharowski, Patrick Meybohm.

**Data curation:** Linda Tanner, Simone Lindau, Kira Berg, Patrick Meybohm.

**Formal analysis:** Linda Tanner, Dirk Hasenclever.

**Funding acquisition:** Kai Zacharowski, Patrick Meybohm.

**Investigation:** Linda Tanner, Kira Berg, Patrick Meybohm.

**Methodology:** Linda Tanner, Dirk Hasenclever.

**Project administration:** Simone Lindau, Kai Zacharowski, Patrick Meybohm.

**Supervision:** Simone Lindau, Patrick Meybohm.

**Writing – original draft:** Linda Tanner, Suma Choorapoikayil.

**Writing – review & editing:** Simone Lindau, Markus Velten, Tobias Schlesinger, Maria Wittmann, Peter Kranke, Kira Berg, Florian Piekarski, Christoph Füllenbach, Suma Choorapoikayil, Dirk Hasenclever, Kai Zacharowski, Patrick Meybohm.

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
