## [Decision Letter · Decision Letter 0]

24 Jul 2020

PONE-D-20-13621

Factors Influencing the Bias between Blood Gas Analysis versus Central Laboratory Hemoglobin Testing. A secondary analysis of a randomized controlled trial.

PLOS ONE

Dear Dr. Meybohm,

Thank you for submitting your manuscript to PLOS ONE. After careful consideration, we feel that it has merit but does not fully meet PLOS ONE’s publication criteria as it currently stands. Therefore, we invite you to submit a revised version of the manuscript that addresses the points raised during the review process.

As pointed out by all reviewers, the topic is interesting, but the study suffers from several limitations. In particular, I strongly believe that professional statistical assistance should be considered. Moreover, clinical implications of your results should be discussed in depth. 

We look forward to receiving your revised manuscript.

Kind regards,

Laura Pasin

Academic Editor

PLOS ONE

Journal Requirements:

Reviewers' comments:

Reviewer's Responses to Questions

**Comments to the Author**

1. Is the manuscript technically sound, and do the data support the conclusions?

Reviewer #1: Partly

Reviewer #2: No

Reviewer #3: Partly

Reviewer #4: Partly

2. Has the statistical analysis been performed appropriately and rigorously? 

Reviewer #1: Yes

Reviewer #2: No

Reviewer #3: Yes

Reviewer #4: No

3. Have the authors made all data underlying the findings in their manuscript fully available?

Reviewer #1: Yes

Reviewer #2: No

Reviewer #3: Yes

Reviewer #4: Yes

4. Is the manuscript presented in an intelligible fashion and written in standard English?

Reviewer #1: Yes

Reviewer #2: Yes

Reviewer #3: Yes

Reviewer #4: Yes

5. Review Comments to the Author

Reviewer #1: • In the introduction, the authors stated: “hemoglobin (Hb) and/or hematocrit levels are one key factor to determine the need for RBC transfusion”. However, many studies have already demonstrated that hemoglobin concentration is not a reliable indicator of transfusion requirement. The authors should add a comment about that.

• While in the abstract the objective of the study is “to investigate the accuracy of BGA hemoglobin testing as compared to CL services”, in the main manuscript the declared aim is different, particularly “to elucidate factors influencing the bias between the most common used methods -CL and BGA- to measure Hb levels in surgical patients.” Which one is the right aim?

• In the study design section, the authors stated that elderly patients scheduled for surgery were included in the trial, but they do not clarify the lower limit of age for inclusion in the study.

• In the result section, the authors stated: “mean bias was reduced by

0.02 g/dl in patients after 24 hours of volume substitution (-0.37 ± 0.29 g/dl) compared to values immediately after surgery (-0.39 ± 0.26 g/dl)”. However, p value is not statistically significant (p=0.202).

• The authors did not discuss the impact of their results on the clinical practice. How could a physician use these results in his daily work when deciding whether to transfuse a patient?

Reviewer #2: The authors set out to study Bias between CL and three different BGA machines. One of them was Co-oximeter. Ideally the study design for this research question would be significantly different from what the authors have used. The design would involve a single blood sample taken per patient and sent for CL and BGA analysis, with stringent control of analytical factors related to specimen collection, transport and processing.

The range of Hb, which is of clinical interest – i.e. where the decisions regarding transfusion are usually made (6 to 8 gm/dl), is poorly represented. Ideally the ranges of Hb should be equally represented <6, 6-8, >8…for example.

I don’t agree with Authors claiming that the bias decreases with lower Hb values. Firstly, there is no explanation provided that why this might be true and secondly, as mentioned before the low range is poorly represented.

The authors then go on to analyse the factors that impact the bias. They have included several factors (patient related and treatment related), without explanation or rationale as to why they included these factors. This exercise then appears to be similar to ‘data mining’ exercise rather than carefully thought of hypothesis generation and testing.

Authors have not taken into consideration the fact that one patient is represented in the data more than once. This fact is important when analysing patient related factors. This is also important as the three centres were very different. For example the Lib07 centre had 6 times more specimens than the other centres. However, the number of patients of the Lib07 and Lib26 combined were only 1/3rd of the entire sample size (result section 2nd Para). This means the number of samples per patient were much more in Lib 07 than others.

I am concerned that the authors have found incidental findings in a poor study design, which are further poorly substantiated in the discussion section. The Authors do talk about smokers and MetHb. The Cooximeter should be able to pick this up in contrast to other BGL machines assessed. So did this effect the bias in the Lib05 centre?

The Authors have used regression analysis to study several factors. Apart from above criticism of lack of rationale/procedure in selecting factors, there is a lot unknown regarding collinearity of factors, excessive representation of one patient etc. Was the assumption of normality met?

The authors have discussed about bias, but the biases appear to be minimal which makes you wonder if its all clinically meaningful. Also authors have used statements such as “Mean bias was reduced by 0.02 g/dl in pts after 24 hours of volume substitution compared to values after surgery” (In results section). Basically there is no difference given the P value and insignificant difference. Hence commenting as the bias is reduced, is incorrect.

The authors took “results within two hours” to define a pair. The actual time of sampling could be 20 minutes to 60 minutes prior to results.

How did they figure out that there was an intervention between the sampling for BGA and CL? How did they manage these pairs where there was an intervention?

We cant compare venous and arterial blood. Authors do acknowledge that in their limitations. I am surprised by their use of word “might”. It definitely affects the bias. We don’t know what was the extent of this confounding factor.

Overall this analysis should be a small part of the Liberal trial results, rather than a full original text publication.

Reviewer #3: In this manuscript, dr Meybohm and colleagues present results of a secondary analysis of an ongoing mRCT on management of blood transfusion after surgery. The aim of their study is to compare Hb levels measured with BGA to Hb levels obtained through central laboratory analysis. They identified several factors that could potentially bias the results of HB analysis by BGA versus CL analysis.

As BGA analysis is frequently used in clinical practice to assess Hb status and potentially trigger clinical decisions, the results of this study may be potentially relevant for several clinicians.

I have a few comments for the Authors which I hope will help them to improve their manuscript:

1. In my opinion, the Authors should better detail in the Discussion the clinical relevance of their findings, and how the results of their study may affect clinical practice and the ongoing LIBERAl trial. E.g. is a bias of 0.09 g/dL clinically relevant?

2. The Authors should better underline in the introduction the difference between their work and previously published studies on difference between BGA and CL analysis. What does this study add to current knowlegde?

3. I understand that blood samples for BGA and CL analysis were taken at different time-points. However, this is not entirely clear from the Methods. I suggest to describe the blood sampling timing in a specific paragraph "blood sampling"

4. Do the Authors have data on BGA and CL agreement during massive transfusion/blood loss? This would be a interesting subgroup analysis

5. The Authors frequently refer to "volume therapy". However, it is unclear what does this mean and how was assessed. Do the Authors refer to maintenance fluids, volume resuscitation, fluid balance? How were these assessed/managed?

6. Please move the list of medications included in the analysis in the Methods section rather than in the Results section

7. The sentence "The LOWESS curve showed a tendency of the bias with decreasing Hb levels." is unclear, please edit.

8. Do the Authors have data on bias of BGA vs CL assessed before, during, and after surgery?

Reviewer #4: This is an interesting subanalysis of the LIBERAL trial, and the restriction to centres with a large number of cases is a sensible one here in the first instance - one does of course want to know whether case volume actually affects accuracy and this could be usefully explored.

What we have here is a number of paired data points by 2 different methods, but also >1 point per patient. So there is clustering here as well and the pairs are not independent. The statistical methodology needs to allow for this but nowhere is this multiplicity of measurements and how it is handled described. This is particularly important in the case of the predictors of bias.

In Figure 3 it would be interesting to look at percentage bias as well here just to see if the trend here actually represents a different type of measurement error. The issue of whether variance depends on value is also interesting and could usefully be explored.

6. PLOS authors have the option to publish the peer review history of their article (what does this mean?). If published, this will include your full peer review and any attached files.

Reviewer #1: No

Reviewer #2: **Yes: **Shivesh Prakash

Reviewer #3: No

Reviewer #4: No

---

## [Author Response · Author response to Decision Letter 0]

20 Aug 2020

All comments have been addressed in the Rebuttal Letter. We thank the editor and reviewers for their helpful remarks.

---

## [Decision Letter · Decision Letter 1]

2 Oct 2020

Factors Influencing the Bias between Blood Gas Analysis versus Central Laboratory Hemoglobin Testing. A secondary analysis of a randomized controlled trial.

PONE-D-20-13621R1

Dear Dr. Meybohm,

We’re pleased to inform you that your manuscript has been judged scientifically suitable for publication and will be formally accepted for publication once it meets all outstanding technical requirements.

Kind regards,

Laura Pasin

Academic Editor

PLOS ONE

Additional Editor Comments (optional):

Reviewers' comments:

Reviewer's Responses to Questions

**Comments to the Author**

1. If the authors have adequately addressed your comments raised in a previous round of review and you feel that this manuscript is now acceptable for publication, you may indicate that here to bypass the “Comments to the Author” section, enter your conflict of interest statement in the “Confidential to Editor” section, and submit your "Accept" recommendation.

Reviewer #1: All comments have been addressed

Reviewer #4: All comments have been addressed

2. Is the manuscript technically sound, and do the data support the conclusions?

Reviewer #1: Yes

Reviewer #4: (No Response)

3. Has the statistical analysis been performed appropriately and rigorously? 

Reviewer #1: Yes

Reviewer #4: (No Response)

4. Have the authors made all data underlying the findings in their manuscript fully available?

Reviewer #1: Yes

Reviewer #4: (No Response)

5. Is the manuscript presented in an intelligible fashion and written in standard English?

Reviewer #1: Yes

Reviewer #4: (No Response)

6. Review Comments to the Author

Reviewer #1: The authors have well responded to the requests.

The findings are presented in a readable way with nuanced conclusions way, and the limitations are well addressed.

I have no further comments.

Reviewer #4: (No Response)

7. PLOS authors have the option to publish the peer review history of their article (what does this mean?). If published, this will include your full peer review and any attached files.

Reviewer #1: No

Reviewer #4: No

---

## [Editor Report · Acceptance letter]

12 Oct 2020

PONE-D-20-13621R1 

Factors Influencing the Bias between Blood Gas Analysis versus Central Laboratory Hemoglobin Testing. A secondary analysis of a randomized controlled trial. 

Dear Dr. Meybohm:

I'm pleased to inform you that your manuscript has been deemed suitable for publication in PLOS ONE. Congratulations! Your manuscript is now with our production department. 

Kind regards, 

on behalf of

Dr. Laura Pasin 

Academic Editor

PLOS ONE